# Factorized Asymptotic Bayesian Inference for Latent Feature Models

**Kohei Hayashi**[*][†]
∗National Institute of Informatics
†JST, ERATO, Kawarabayashi Large Graph Project
kohei-h@nii.ac.jp

**Ryohei Fujimaki**
NEC Laboratories America
rfujimaki@nec-labs.com

## Abstract

This paper extends factorized asymptotic Bayesian (FAB) inference for latent feature models (LFMs). FAB inference has not been applicable to models, including LFMs, without a specific condition on the Hessian matrix of a complete log-likelihood, which is required to derive a "factorized information criterion" (FIC). Our asymptotic analysis of the Hessian matrix of LFMs shows that FIC of LFMs has the same form as those of mixture models. FAB/LFMs have several desirable properties (e.g., automatic hidden states selection and parameter identifiability) and empirically perform better than state-of-the-art Indian Buffet processes in terms of model selection, prediction, and computational efficiency.

## 1   Introduction

Factorized asymptotic Bayesian (FAB) inference is a recently-developed Bayesian approximation inference method for model selection of latent variable models [5, 6]. FAB inference maximizes a computationally tractable lower bound of a "factorized information criterion" (FIC) which converges to a marginal log-likelihood for a large sample limit. In application with respect to mixture models (MMs) and hidden Markov models, previous work has shown that FAB inference achieves as good or even better model selection accuracy as state-of-the-art non-parametric Bayesian (NPB) methods and variational Bayesian (VB) methods with less computational cost. One of the interesting characteristics of FAB inference is that it estimates both models (e.g., the number of mixed components for MMs) and parameter values without priors (i.e., it asymptotically ignores priors), and it does not have a hand-tunable hyper-parameter. With respect to the trade-off between controllability and automation, FAB inference places more importance on automation.

Although FAB inference is a promising model selection method, as yet it has only been applicable to models satisfying a specific condition that the Hessian matrix of a complete log-likelihood (i.e., of a log-likelihood over both observed and latent variables) must be block diagonal, with only a part of the observed samples contributing individual sub-blocks. Such models include basic latent variable models as MMs [6]. The application of FAB inference to more advanced models that do not satisfy the condition remains to be accomplished.

This paper extends an FAB framework to latent feature models (LFMs) [9, 17]. Model selection for LFMs (i.e., determination of the dimensionality of latent features) has been addressed by NBP and VB methods [10, 3]. Although they have shown promising performance in such applications as link prediction [16], their high computational costs restrict their applications to large-scale data.

Our asymptotic analysis of the Hessian matrix of the log-likelihood shows that FICs for LFMs have the same form as those for MMs, despite the fact that LFMs do not satisfy the condition explained above (see Lemma 1). Eventually, as FAB/MMs, FAB/LFMs offer several desirable properties, such as FIC convergence to a marginal log-likelihood, automatic hidden states selection, and monotonic increase in the lower FIC bound through iterative optimization. Further we conduct two analysis in

Section 3: 1) we relate FAB E-steps to a convex concave procedure (CCCP) [29]. Inspired by this analysis, we propose a shrinkage acceleration method which drastically reduces computational cost in practice, and 2) we show that FAB/LFMs have parameter identifiability. This analysis offers a natural guide to the merging post-processing of latent features. Rigorous proofs and assumptions with respect to the main results are given in the supplementary materials.

**Notation** In this paper, we denote the $(i, j)$-th element, the $i$-th row vector, and the $j$-th column vector of $\mathbf{A}$ by $a_{ij}$, $\mathbf{a}_i$, and $\mathbf{a}_{\cdot j}$, respectively.

## 1.1 Related Work

**FIC for MMs** Suppose we have $N \times D$ observed data $\mathbf{X}$ and $N \times K$ latent variables $\mathbf{Z}$. FIC considers the following alternative representation of the marginal log-likelihood:

$$\log p(\mathbf{X}|\mathcal{M}) = \max_q \left\{ \sum_{\mathbf{Z}} q(\mathbf{Z}) \log \frac{p(\mathbf{X}, \mathbf{Z}|\mathcal{M})}{q(\mathbf{Z})} \right\}, \ p(\mathbf{X}, \mathbf{Z}|\mathcal{M}) = \int p(\mathbf{X}, \mathbf{Z}|\mathcal{P}) p(\mathcal{P}|\mathcal{M}) \mathrm{d}\mathcal{P}, \quad (1)$$

where $q(\mathbf{Z})$ is a variational distribution on $\mathbf{Z}$; $\mathcal{M}$ and $\mathcal{P}$ are a model and its parameter, respectively. In the case of MMs, $\log p(\mathbf{X}, \mathbf{Z}|\mathcal{P})$ can be factorized into $\log p(\mathbf{Z})$ and $\log p(\mathbf{X}|\mathbf{Z}) = \sum_k \log p_k(\mathbf{X}|\mathbf{z}_{\cdot k})$, where $p_k$ is the $k$-th observation distribution (we here omit parameters for notational simplicity.) We can then approximate $p(\mathbf{X}, \mathbf{Z}|\mathcal{M})$ by individually applying Laplace's method [28] to $\log p(\mathbf{Z})$ and $\log p_k(\mathbf{X}|\mathbf{z}_{\cdot k})$:

$$p(\mathbf{X}, \mathbf{Z}|\mathcal{M}) \approx p(\mathbf{X}, \mathbf{Z}|\hat{\mathcal{P}}) \frac{(2\pi)^{D_Z/2}}{N^{D_Z/2} \det |\mathbf{F}_Z|^{1/2}} \prod_{k=1}^{K} \frac{(2\pi)^{D_k/2}}{(\sum_n z_{nk})^{D_k/2} \det |\mathbf{F}_k|^{1/2}}, \quad (2)$$

where $\hat{\mathcal{P}}$ is the maximum likelihood estimator (MLE) of $p(\mathbf{X}, \mathbf{Z}|\mathcal{P})$.[1] $D_Z$ and $D_k$ are the parameter dimensionalities of $p(\mathbf{Z})$ and $p_k(\mathbf{X}|\mathbf{z}_{\cdot k})$, respectively. $\mathbf{F}_Z$ and $\mathbf{F}_k$ are $-\nabla\nabla \log p(\mathbf{Z})|_{\hat{\mathcal{P}}}/N$ and $-\nabla\nabla \log p_k(\mathbf{X}|\mathbf{z}_{\cdot k})|_{\hat{\mathcal{P}}}/(\sum_n z_{nk})$, respectively. Under conditions for asymptotic ignoring of $\log \det |\mathbf{F}_Z|$ and $\log \det |\mathbf{F}_k|$, substituting Eq.(2) into (1) gives the FIC for MMs as follows:

$$\mathrm{FIC}_{\mathrm{MM}} \equiv \max_q \mathbb{E}_q \left[ \log p(\mathbf{X}, \mathbf{Z}|\hat{\mathcal{P}}) - \frac{D_Z}{2} \log N - \sum_k \frac{D_k}{2} \log \sum_n z_{nk} \right] + H(q), \quad (3)$$

where $H(q)$ is the entropy of $q(\mathbf{Z})$. The most important term in $\mathrm{FIC}_{\mathrm{MM}}$ (3) is $\log(\sum_n z_{nk})$, which offers such theoretically desirable properties for FAB inference as automatic shrinkage of irrelevant latent variables and parameter identifiability [6].

Direct optimization of $\mathrm{FIC}_{\mathrm{MM}}$ is difficult because: **(i)** evaluation of $\mathbb{E}_q[\log \sum_n z_{nk}]$ is computationally infeasible, and **(ii)** the MLE is not available in principle. Instead, FAB optimizes a tractable lower bound of an FIC [6]. For **(i)**, since $-\log \sum_n z_{nk}$ is a convex function, its linear approximation at $N\tilde{\pi}_k > 0$ yields the lower bound:

$$-\sum_k \frac{D_k}{2} \mathbb{E}_q \left[ \log \sum_n z_{nk} \right] \geq -\sum_k \frac{D_k}{2} \left( \log N\tilde{\pi}_k + \frac{\sum_n \mathbb{E}_q[z_{nk}]/N - \tilde{\pi}_k}{\tilde{\pi}_k} \right), \quad (4)$$

where $0 < \tilde{\pi}_k \leq 1$ is a linearization parameter. For **(ii)**, since, from the definition of the MLE, the inequality $\log p(\mathbf{X}, \mathbf{Z}|\hat{\mathcal{P}}) \geq \log p(\mathbf{X}, \mathbf{Z}|\mathcal{P})$ holds for any $\mathcal{P}$, we optimize $\mathcal{P}$ along with $q$. Alternating maximization of the lower bound with respect to $q$, $\mathcal{P}$, and $\tilde{\boldsymbol{\pi}}$ guarantees a monotonic increase in the FIC lower bound [6].

**Infinite LFMs and Indian Buffet Process** The IBP [10, 11] is a nonparametric prior over infinite LFMs. It enables us to express an infinite number of latent features, and making it possible to adjust model complexity on the basis of observations. Infinite IBPs have still been actively studied in terms of both applications (e.g., link prediction [16]) and model representations (e.g., latent attribute models [19]). Since naive Gibbs sampling requires unrealistic computational cost, acceleration algorithms such as accelerated sampling [2] and VB [3] have been developed. Reed and Ghahramani [22] have recently proposed an efficient MAP estimation framework of an IBP model via submodular optimization, which is referred to as maximum-expectation IBP (MEIBP). As similar to FIC, "MAD-Bayes" [1] considers asymptotics of MMs and LFMs, but it is based on a limiting case that the noise variance goes to zero, which yields a prior-derived regularization term.

## 2 FIC and FAB Algorithm for LFMs

LFMs assume underlying relationships for $\mathbf{X}$ with binary features $\mathbf{Z} \in \{0,1\}^{N \times K}$ and linear bases $\mathbf{W} \in \mathbb{R}^{D \times K}$ such that, for $n = 1, \ldots, N$,

$$\mathbf{x}_n = \mathbf{W}\mathbf{z}_n + \mathbf{b} + \boldsymbol{\varepsilon}_n, \tag{5}$$

where $\boldsymbol{\varepsilon}_n \sim N(\mathbf{0}, \boldsymbol{\Lambda}^{-1})$ is the Gaussian noise having the diagonal precision matrix $\boldsymbol{\Lambda} \equiv \mathrm{diag}(\boldsymbol{\lambda})$, and $\mathbf{b} \in \mathbb{R}^D$ is a bias term. For later convenience, we define the centered observation $\bar{\mathbf{X}} = \mathbf{X} - \mathbf{1}\mathbf{b}^\top$. $\mathbf{Z}$ follows a Bernoulli prior distribution $z_{nk} \sim \mathrm{Bern}(\pi_k)$ with a mean parameter $\pi_k$. The parameter set $\mathcal{P}$ is defined as $\mathcal{P} \equiv \{\mathbf{W}, \mathbf{b}, \boldsymbol{\lambda}, \boldsymbol{\pi}\}$. Also, we denote parameters with respect to the $d$-th dimension as $\boldsymbol{\theta}_d = (\mathbf{w}_d, b_d, \lambda_d)$. Similarly with other FAB frameworks, the log-priors of $\mathcal{P}$ are assumed to be constant with respect to $N$, i.e., $\lim_{N \to \infty} \frac{\log p(\mathcal{P}|\mathcal{M})}{N} = 0$

In the case of MMs, we implicitly use the fact that: **A1**) parameters of $p_k(\mathbf{X}|\mathbf{z}_{\cdot k})$ are mutually independent for $k = 1, \ldots, K$ (in other words, $\nabla\nabla \log p(\mathbf{X}|\mathbf{Z})$ is block diagonal having $K$ blocks), and **A2**) the number of observations which contribute $\nabla\nabla \log p_k(\mathbf{X}|\mathbf{z}_{\cdot k})$ is $\sum_n z_{nk}$. These conditions naturally yield the FAB regularization term $\log \sum_n z_{nk}$ by the Laplace approximation of MMs (2). However, since $\boldsymbol{\theta}_d$ is shared by all latent features in LFMs, **A1** and **A2** are not satisfied. In the next section, we address this issue and derive FIC for LFMs.

### 2.1 FICs for LFMs

The following lemma plays the most important role in our derivation of FICs for LFMs.

**Lemma 1.** *Let $\mathbf{F}^{(d)}$ be the Hessian matrix of the negated log-likelihood with respect to $\boldsymbol{\theta}_d$, i.e., $-\nabla\nabla \log p(\mathbf{x}_{\cdot d}|\mathbf{Z}, \boldsymbol{\theta}_d)$. Under some mild assumptions (see the supplementary materials), the following equality holds:*

$$\log\det|\mathbf{F}^{(d)}| = \sum_k \log \frac{\sum_n z_{nk}}{N} + O_p(1). \tag{6}$$

An important fact is that the $\log \sum_n z_{nk}$ term naturally appears in $\log\det|\mathbf{F}^{(d)}|$ without **A1** and **A2**. Lemma 1 induces the following theorem, which states an asymptotic approximation of a marginal complete log-likelihood, $\log p(\mathbf{X}, \mathbf{Z}|\mathcal{M})$.

**Theorem 2.** *If Lemma 1 holds and the joint marginal log-likelihood is bounded for a sufficiently large $N$, it can be asymptotically approximated as:*

$$\log p(\mathbf{X}, \mathbf{Z}|\mathcal{M}) = J(\mathbf{Z}, \hat{\mathcal{P}}) + O_p(1), \tag{7}$$

$$J(\mathbf{Z}, \mathcal{P}) \equiv \log p(\mathbf{X}, \mathbf{Z}|\mathcal{P}) - \frac{|\mathcal{P}| - DK}{2} \log N - \frac{D}{2} \sum_k \log \sum_n z_{nk}. \tag{8}$$

It is worth noting that, if we evaluate the model complexity of $\boldsymbol{\theta}_d$ ($\log\det|\mathbf{F}^{(d)}|$) by $N$, i.e., if we apply Laplace's method without Lemma 1, Eq. (7) falls into Bayesian Information Criterion [23], which tells us that the model complexity relevant to $\boldsymbol{\theta}_d$ increases not $O(K \log N)$ but $O(\sum_k \log \sum_n z_{nk})$.

By substituting approximation (7) into Eq. (1), we obtain the FIC of the LFM as follows:

$$\mathrm{FIC}_{\mathrm{LFM}} \equiv \max_q \mathbb{E}_q[J(\mathbf{Z}, \hat{\mathcal{P}})] + H(q). \tag{9}$$

It is interesting that $\mathrm{FIC}_{\mathrm{LFM}}$ (9) and $\mathrm{FIC}_{\mathrm{MM}}$ (3) have exactly the same representation despite the fact that LFMs do not satisfy **A1** and **A2**. This indicates the wide applicability of FICs and suggests that FIC representation of approximated marginal log-likelihoods is feasible not only for MMs but also for more general (discrete) latent variable models.

Since the asymptotic constant terms of Eq. (7) are not affected by the expectation of $q(\mathbf{Z})$, the difference between the FIC and the marginal log-likelihood is asymptotically constant; in other words, the distance between $\log p(\mathbf{X}|\mathcal{M})/N$ and $\mathrm{FIC}_{\mathrm{LFM}}/N$ is asymptotically small.

**Corollary 3.** *For $N \to \infty$, $\log p(\mathbf{X}|\mathcal{M}) = \mathrm{FIC}_{\mathrm{LFM}} + O_p(1)$ holds.*

## 2.2 FAB/LFM Algorithm

As with the case of MMs (3), $FIC_{LFM}$ is not available in practice, and we employ the lower bounding techniques **(i)** and **(ii)**. For LFMs, we further introduce a mean-filed approximation on $\mathbf{Z}$, i.e., we restrict the class of $q(\mathbf{z}_n)$ to a factorized form: $q(\mathbf{z}_n) = \prod_k \tilde{q}(z_{nk}|\mu_{nk})$, where $\tilde{q}(z|\mu)$ is a Bernoulli distribution with a mean parameter $\mu = \mathbb{E}_q[z]$. Rather than this approximation's making the FIC lower bound looser (the equality (1) no longer holds), the variational distribution has a closed-form solution. Note that this approximation does not cause significant performance degradation in VB contexts [20, 25]. The VB-extension of IBP [3] also uses this factorized assumption.

By applying **(i)**, **(ii)**, and the mean-field approximation, we obtain the lower bound: $\mathcal{L}(q, \mathcal{P}, \tilde{\boldsymbol{\pi}}) =$

$$\mathbb{E}_q\left[\log p(\mathbf{X}|\mathbf{Z}, \boldsymbol{\Theta}) + \log p(\mathbf{Z}|\boldsymbol{\pi}) + \text{RHS of (4)}\right] - \frac{2D+K}{2}\log N + \sum_n H(q(\mathbf{z}_n)). \tag{10}$$

An FAB algorithm alternatingly maximizes $\mathcal{L}(q, \mathcal{P}, \tilde{\boldsymbol{\pi}})$ with respect to $\{\{\boldsymbol{\mu}_n\}, \mathcal{P}, \tilde{\boldsymbol{\pi}}\}$. Notice that the algorithm described below monotonically increases $\mathcal{L}$ in every single step, and therefore we are guaranteed to obtain a local maximum. This monotonic increase in $\mathcal{L}$ gives us a natural stopping condition with a tolerance $\delta$: if $(\mathcal{L}^t - \mathcal{L}^{t-1})/N < \delta$ then stop the algorithm, where we denote the value of $\mathcal{L}$ at the $t$-th iteration by $\mathcal{L}^t$.

**FAB E-step**  In the FAB E-step, we update $\mu_n$ in a way similar to that with the variational mean-field inference in a restricted Boltzmann machine [20]. Taking the gradient of $\mathcal{L}$ with respect to $\boldsymbol{\mu}_n$ and setting it to zero yields the following fixed-point equations:

$$\mu_{nk} = g\left(c_{nk} + \eta(\pi_k) - D/2N\tilde{\pi}_k\right) \tag{11}$$

where $g(x) = (1 + \exp(-x))^{-1}$ is the sigmoid function, $c_{nk} = \mathbf{w}_{\cdot k}^\top \boldsymbol{\Lambda}(\bar{\mathbf{x}}_n - \sum_{l \neq k} \mu_{nl} \mathbf{w}_{\cdot l} - \frac{1}{2}\mathbf{w}_{\cdot k})$, and $\eta(\pi_k) = \log \frac{\pi_k}{1-\pi_k}$ is a natural parameter of the prior of $\mathbf{z}_{\cdot k}$. Update equation (11) is a form of coordinate descent, and every update is guaranteed to increase the lower bound [25]. After several iterations of Eq. (11) over $k = 1, \ldots, K$, we are able to obtain a local maximum of $\mathbb{E}_q[\mathbf{z}_n] = \boldsymbol{\mu}_n$ and $\mathbb{E}_q[\mathbf{z}_n \mathbf{z}_n^\top] = \boldsymbol{\mu}_n \boldsymbol{\mu}_n^\top + \text{diag}(\boldsymbol{\mu}_n - \boldsymbol{\mu}_n^2)$.

One unique term in Eq. (11) is $-\frac{D}{2N\tilde{\pi}_k}$, which originated in the $\log \sum_n z_{nk}$ term in Eq. (8). In updating $\mu_{nk}$ (11), the smaller $\tilde{\pi}_k$ (or equivalent to $\pi_k$ by Eq. (12)) is, the smaller $\mu_{nk}$ is. And a smaller $\mu_{nk}$ is likely to induce a smaller $\tilde{\pi}_k$ (see Eq. (12)). This results in the shrinking of irrelevant features, and therefore FAB/LFMs are capable of automatically selecting feature dimensionality $K$. This regularization effect is induced independently of prior (i.e., asymptotic ignorance of prior) and is known as "model induced regularization" which is caused by Bayesian marginalization in singular models [18]. Notice that Eq. (11) offers another shrinking effect, by means of $\eta(\pi_k)$, which is a prior-based regularization. We empirically show that the latter shrinking effect is too weak to mitigate over-fitting and the FAB algorithm achieves faster convergence, with respect to $N$, to the true model (see Section 4.) Note that if we only use the effect of $\eta(\pi_k)$ (i.e. setting $D/2N\tilde{\pi}_k = 0$), then update equation (11) is equivalent to that of variational EM.

**FAB M-step**  The FAB M-step is equivalent to the M-step in the EM algorithm of LFMs; the solutions of $\mathbf{W}, \boldsymbol{\Lambda}$ and $\mathbf{b}$ are given as in closed form and is exactly the same as those of PPCA [24] (see the supplementary materials.) For $\tilde{\boldsymbol{\pi}}$ and $\boldsymbol{\pi}$, we obtain the following solutions:

$$\pi_k = \tilde{\pi}_k = \sum_n \mu_{nk}/N. \tag{12}$$

**Shrinkage step**  As we have explained, in principle, the FAB regularization term $\frac{D}{2N\tilde{\pi}_k}$ in Eq. (11) automatically eliminates irrelevant latent features. While the elimination does not change the value of $\mathbb{E}_q[\log(\mathbf{X}|\mathbf{Z}, \mathcal{P})]$, removing them from the model increases $\mathcal{L}$ due to a decrease in model complexity. We eliminate shrunken features after FAB E-step in terms of that LFMs approximate $\mathbf{X}$ by $\sum_k \boldsymbol{\mu}_{\cdot k}\mathbf{w}_{\cdot k}^\top + \mathbf{1}\mathbf{b}^\top$. When $\sum_n \mu_{nk}/N = 0$, the $k$-th feature does not affect to the approximation ($\sum_l \mathbf{z}_{\cdot l}\mathbf{w}_{\cdot l}^\top = \sum_{l \neq k}\mathbf{z}_{\cdot l}\mathbf{w}_{\cdot l}^\top$), and we simply remove it. When $\sum_n \mu_{nk}/N = 1$, $\mathbf{w}_k$ can be seen as a bias ($\sum_l \mathbf{z}_{\cdot l}\mathbf{w}_{\cdot l}^\top = \sum_{l \neq k}\mathbf{z}_{\cdot l}\mathbf{w}_{\cdot l}^\top + \mathbf{1}\mathbf{w}_{\cdot k}^\top$), and we update $\mathbf{b}^{\text{new}} = \mathbf{b} + \mathbf{w}_k$ and then remove it.

**Algorithm 1** The FAB algorithm for LFMs.

1: Initialize $\{\boldsymbol{\mu}_n\}$
2: **while** Convergence **do**
3:     Update $\mathcal{P}$
4:     `accelerateShrinkage`$(\{\boldsymbol{\mu}_n\})$
5:     **for** $k = 1, \ldots, K$ **do**
6:         Update $\{\mu_{nk}\}$ by Eq. (11)
7:     **end for**
8:     Shrink unnecessary latent features
9:     **if** $(\mathcal{L}^t - \mathcal{L}^{t-1})/N < \delta$ **then**
10:        $\{\{\boldsymbol{\mu}'_n\}, \mathbf{W}'\} \leftarrow \texttt{merge}(\{\boldsymbol{\mu}_n\}, \mathbf{W})$
11:        **if** $\dim(\mathbf{W}') = \dim(\mathbf{W})$ **then** Converge
12:        **else** $\{\boldsymbol{\mu}_n\} \leftarrow \{\boldsymbol{\mu}'_n\}, \mathbf{W} \leftarrow \mathbf{W}'$
13:     **end if**
14: **end while**

---

**Algorithm 2** `accelerateShrinkage`

**input** $\{\boldsymbol{\mu}_n\}$
1: **for** $k = 1, \ldots, K$ **do**
2:     $\mathbf{c}_k \leftarrow (\bar{\mathbf{X}} - \sum_{l \neq k} \boldsymbol{\mu}_{\cdot l} \mathbf{w}_{\cdot l}^\top - \frac{1}{2} \mathbf{1} \mathbf{w}_{\cdot k}^\top) \boldsymbol{\Lambda} \mathbf{w}_{\cdot k}$
3:     **for** $t = 1, \ldots, T_{\text{shrink}}$ **do**
4:         Update $\{\mu_{nk}\}$ by Eq. (11)
5:         Update $\boldsymbol{\pi}$ and $\tilde{\boldsymbol{\pi}}$ by Eq. (12)
6:     **end for**
7: **end for**

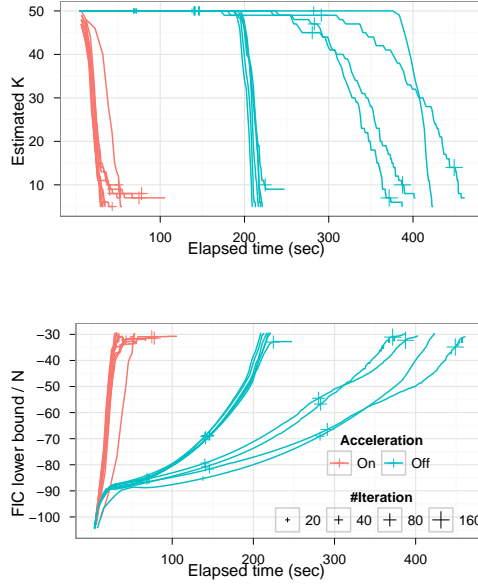

Figure 1: Time evolution of $K$ (top) and $\mathcal{L}/N$ (bottom) in *FAB* with and without shrinkage acceleration ($D = 50$ and $K = 5$). Different lines represent different random starts.

This model shrinkage also works to avoid the ill-conditioning of the FIC; if there are latent features that are never activated ($\sum_n \mu_{nk}/N = 0$) or always activated ($\sum_n \mu_{nk}/N = 1$), the FIC will no longer be an approximation of the marginal log-likelihood. Algorithm 1 summarizes whole procedures with respect to the FAB/LFMs. Note that details regarding sub-routines `accelerateShrinkage()` and `merge()` are explained in Section 3.

## 3 Analysis and Refinements

**CCCP Interpretation and Shrinkage Acceleration** Here we interpret the alternating updates of $\boldsymbol{\mu}$ and $\tilde{\boldsymbol{\pi}}$ as a convex concave procedure (CCCP) [29] and consider to eliminate irrelevant features in early steps to reduce computational cost. By substituting an optimality condition $\tilde{\pi}_k = \sum_n \mu_{nk}/N$ (12) into the lower bound, we obtain

$$\mathcal{L}(q) = -\frac{D}{2} \sum_k \log \sum_n \mu_{nk} + \left( \sum_n (\mathbf{c}_n + \boldsymbol{\eta})^\top \boldsymbol{\mu}_n + H(q) \right) + \text{const.} \qquad (13)$$

The first and second terms are convex and concave with respect to $\mu_{nk}$, respectively. The CCCP solves Eq.(13) by iteratively linearizing the first term around $\mu_{nk}^{t-1}$. By setting the derivative of the "linearized" objective to be zero, we obtain the CCCP update as follows:

$$\mu_{nk}^t = g\left( c_{nk} + \eta(\pi_k) - \frac{D}{2} \sum_n \mu_{nk}^{t-1} \right). \qquad (14)$$

By taking $N\tilde{\pi}_k = \sum_n \mu_{nk}^{t-1}$ into account, Eq.(14) is equivalent to Eq.(11).

This new view of the FAB optimization gives us an important insight to accelerate the algorithm. By considering the FAB optimization as the alternating maximization in terms of $\mathcal{P}$ and $\boldsymbol{\mu}$ ($\tilde{\pi}$ is removed), it is natural to take multiple CCCP steps (14). Such multiple CCCP steps in each FAB-EM step is expected to accelerate the shrinkage effect discussed in the previous section because the

regularization in terms of $-D/2(\sum_n \mu_{nk})$ causes the effect. Eventually, it is expected to reduce the total computational cost since we may be able to remove irrelevant latent features in earlier iterations. We summarize the whole routine of `accelerateShrinkage()` based on the CCCP in Algorithm 2. Note that, in practice, we update $\boldsymbol{\pi}$ along with $\tilde{\boldsymbol{\pi}}$ for further acceleration of the shrinkage. We empirically confirmed that Algorithm 2 significantly reduced computational costs (see Section 4 and Figure 1.) Further discussion of this this update (an exponentiated gradient descent interpretation) can be found in the supplementary materials.

**Identifiability and Merge Post-processing**   Parameter identifiability is an important theoretical aspect in learning algorithms for latent variable models. It has been known [26, 27] that generalization error significantly worsens if the mapping between parameters and functions is not one-to-one (i.e., is non-identifiable.) Let us consider the LFM case of $K = 2$. If $\mathbf{w}_{\cdot 1} = \mathbf{w}_{\cdot 2}$, then any combination of $\mu_{n1}$ and $\mu_{n2} = 2\mu - \mu_{n1}$ will have the same representation: $\mathbb{E}_q[\mathbb{E}_x[\bar{x}_{nd}|\boldsymbol{\theta}_d]] = w_{d1}(\mu_{n1} + \mu_{n2}) = 2w_{d1}\mu$, and therefore the MLE is non-identifiable.

The following theorem shows that FAB inference resolves such non-identifiability in LFMs.

**Theorem 4.** *Let $\mathcal{P}^*$ and $q^*$ be stationary points of $\mathcal{L}$ such that $0 < \sum_n \mu_{nk}^*/N < 1$ for $k = 1, \ldots, K$ and $|\bar{\mathbf{x}}_n^\top \boldsymbol{\Lambda}^* \mathbf{w}_{\cdot k}^*| < \infty$ for $k = 1, \ldots, K$, $n = 1, \ldots, N$. Then, $\mathbf{w}_{\cdot k}^* = \mathbf{w}_{\cdot l}^*$ is a sufficient condition of $\sum_n \mu_{nk}^*/N = \sum_n \mu_{nl}^*/N$.*

For the ill-conditioned situation described above, the FAB algorithm has a unique solution that balances the sizes of latent features. In large sample limit, both FAB and EM reach the same ML value. The point is, for LFMs, ML solutions are not unique and EM is likely to choose large-$K$-solutions because of this non-identifiability issue. On the other hands, FAB prefers to small-$K$ ML solutions on the basis of the regularizer. In addition, Theorem 4 gives us an important insight about post-processing of latent features. If $\mathbf{w}_{\cdot k}^* = \mathbf{w}_{\cdot l}^*$, then $\mathbb{E}_q[\log p(\mathbf{X}, \mathbf{Z}|\mathcal{M}^*)]$ is equivalent without relation to $\mu_{nk}$ and $\mu_{nl}$, while model complexity is smaller if we only have one latent feature. Therefore, if $\mathbf{w}_{\cdot k}^* = \mathbf{w}_{\cdot l}^*$, merging these two latent features increases $\mathcal{L}$, i.e., $\mathbf{w}_{\cdot k}^* = 2\mathbf{w}_{\cdot k}^*$ and $\boldsymbol{\mu}_{\cdot k}^* = \frac{\boldsymbol{\mu}_{\cdot k}^* + \boldsymbol{\mu}_{\cdot l}^*}{2}$. In practice, we search for such overlapping features on the basis of a Euclidean distance matrix of $\mathbf{W}^*$ and $\mathbf{w}_{\cdot k}^*$ for $k = 1, \ldots, K$ and merge them if the lower bound increases after the post-processing. We empirically found that a few merging operations were likely to occur in real world data sets. The algorithm of `merge()` is summarized in the supplementary materials.

## 4   Experiments

We have evaluated FAB/LFMs in terms of computational speed, model selection accuracy, and prediction performance with respect to missing values. We compared *FAB* inference and the variational *EM* algorithm (see Section 2.2) with an *IBP* that utilized fast Gibbs sampling [2], a *VB* [3] having a finite $K$, and *MEIBP* [22]. *IBP* and *MEIBP* select a model which maximizes posterior probability. For *VB*, we performed inference with $K = 2, \ldots, D$ and selected the model having the highest free energy. *EM* selects $K$ using the shrinkage effect of $\eta$ as we have explained in Section 2.2.

All the methods were implemented in Matlab (for *IBP*, *VB*, and *MEIBP*, we used original codes released by the authors), and the computational performance were fairly compared. For *FAB* and *EM*, we set $\delta = 10^{-4}$ (this was not sensitive) and $T_{\text{shrink}} = 100$ (*FAB* only); $\{\boldsymbol{\mu}_n\}$ were randomly and uniformly initialized by 0 and 1; the initial number of latent features was set to $\min(N, D)$ as well as *MEIBP*. Since the softwares of *IBP*, *VB*, and *MEIBP* did not learn the standard deviation of the noise ($1/\sqrt{\boldsymbol{\lambda}}$ in *FAB*), we fixed it to 1 for artificial simulations, which is the true standard deviation of toy data, and $0.75$ for real data by following the original papers [2, 22]. We set other parameters with software default values. For example, $\alpha$, a hyperparameter of *IBP*, was set to 3, which might cause overestimation of $K$. As common preprocessing, we normalized $\mathbf{X}$ (i.e., the sample variance is 1) in all experiments.

**Artificial Simulations**   We first conducted artificial simulations with fully-observed synthetic data generated by model (5) having a fixed $\lambda_k = 1$ and $\pi_k = 0.5$. Figure 1 shows the results of a comparison between *FAB* with and without shrinkage acceleration.[2] Clearly, our shrinkage acceleration

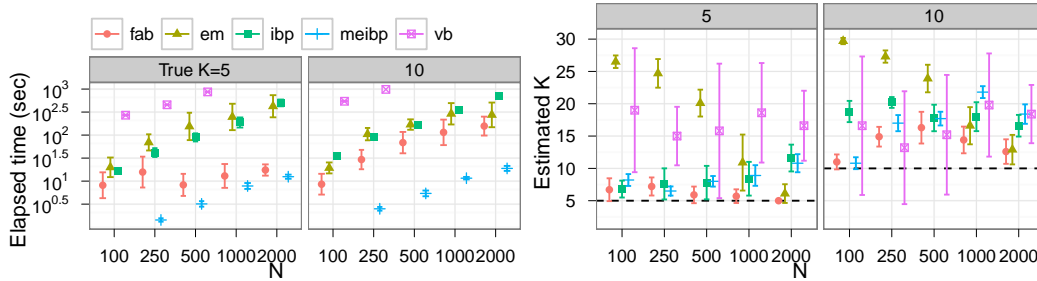

Figure 2: Comparative evaluation of the artificial simulations in terms of $N$ v.s. elapsed time (left) and selected $K$ (right). Each error-bar shows the standard deviation over 10 trials ($D = 30$).

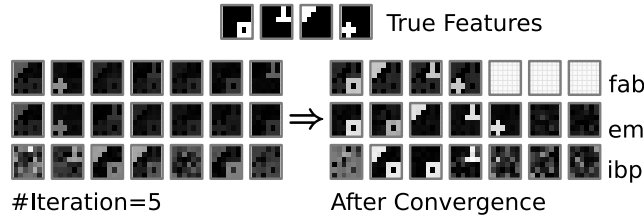

Figure 3: Learned **W**s in `block` data.

significantly reduced computational cost by eliminating irrelevant features in the early steps, while both algorithms achieved roughly the same objective value $\mathcal{L}$ and model selection performance at the convergence. Figure 2 shows the results of a comparison between *FAB* (with acceleration) and the other methods. While *MEIBP* was much faster than *FAB* in terms of elapsed computational time, *FAB* achieved the most accurate estimation of $K$, especially for large $N$.

**Block Data** We next demonstrate performance of FAB/LFMs in terms of learning features. We used the `block` data, a synthetic data originally used in [10]. Observations were generated by combining four distinct patterns (i.e., $K = 4$, see Figure 3) with Gaussian noise, on 6 by 6 pixels (i.e., $D = 36$). We prepared the results of $N = 2000$ samples with the noise standard deviation 0.3 and no missing values (more results can be found in the supplementary materials.) Figure 3 compares estimated features of each method on early learning phase (at the 5th iteration) and after the convergence (the result displayed is the example which has the median log-likelihood over 10 trials.) Note that, we omitted *MEIPB* since we observed that its parameter setting was very sensitive for this data. While *EM* and *IBP* retain irrelevant features, *FAB* successfully extracts the true patterns without irrelevant features.

**Real World Data** We finally evaluated predictive performance by using the real data sets described in Table 1. We randomly removed 30% of data with 5 different random seeds and treated them as missing values, and we measured predictive and training log-likelihood (PLL and TLL) for them. Table 1 summarizes the results with respect to elapsed computational time (hours), selected $K$, PLL, and TLL. Note that, for cases when the computational time for a method exceeded 50 hours, we stopped the program after that iteration.[3] Since *MEIBP* is the method for non-negative data, we omitted the results of those containing negative values. Also, since *MEIBP* did not finish the first iteration within 50 hours for `yaleB` and `USPS` data, we set the initial $K$ as 100. *FAB* consistently achieved good predictive performance (higher PLL) with low computational cost. Although *MEIBP* performed faster than *FAB* with appropriately set the initial value of $K$ (i.e., `yaleB` and `USPS`), PLLs of *FAB* were much better than those of *MEIBP*. In terms of $K$, *FAB* typically achieved a more compact and better model representation than the others (smaller $K$). Another important observation is that *FAB* have much smaller differences between TLL and PLL than the others. This suggests that *FAB*'s unique regularization worked well for mitigating over-fitting. For the large sample data sets (`EEG`, `Piano`, `USPS`), PLLs of *FAB* and *EM* were competitive with one another;

Table 1: Results on real-world data sets. The best result (e.g., the smallest $K$ in model selection) and those not significantly worse than it are highlighted in boldface. We used a one-side $t$-test with 95% confidence. *We exclude the results of *MEIBP* for yaleB and USPS from the $t$-test because of the different experimental settings (initial K was smaller than the others. See the body text for details.)

| Data | Method | Time (h) | $K$ | PLL | TLL |
|---|---|---|---|---|---|
| Sonar [4] | FAB | $< \mathbf{0.01}$ | $\mathbf{4.4 \pm 1.1}$ | $\mathbf{-1.25 \pm 0.02}$ | $-1.14 \pm 0.03$ |
| $208 \times 49$ | EM | $< 0.01$ | $48.8 \pm 0.5$ | $-4.04 \pm 0.46$ | $-0.08 \pm 0.07$ |
| | IBP | $3.3$ | $69.6 \pm 4.8$ | $-4.48 \pm 0.15$ | $\mathbf{0.13 \pm 0.02}$ |
| | MEIBP | $< 0.01$ | $45.4 \pm 1.7$ | $-18.10 \pm 1.90$ | $-15.60 \pm 1.80$ |
| Libras [4] | FAB | $< \mathbf{0.01}$ | $\mathbf{19.0 \pm 0.7}$ | $-0.63 \pm 0.03$ | $-0.42 \pm 0.03$ |
| $360 \times 90$ | EM | $0.01$ | $75.6 \pm 8.6$ | $-0.68 \pm 0.11$ | $\mathbf{0.76 \pm 0.24}$ |
| | IBP | $4.8$ | $36.4 \pm 1.1$ | $\mathbf{-0.18 \pm 0.01}$ | $0.13 \pm 0.01$ |
| | MEIBP | $0.05$ | $40.8 \pm 1.3$ | $-11.30 \pm 2.00$ | $-10.70 \pm 1.80$ |
| Auslan [14] | FAB | $\mathbf{0.04}$ | $\mathbf{6.0 \pm 0.7}$ | $\mathbf{-1.34 \pm 0.15}$ | $-0.92 \pm 0.02$ |
| $16180 \times 22$ | EM | $0.2$ | $22 \pm 0$ | $-1.79 \pm 0.27$ | $-0.78 \pm 0.02$ |
| | IBP | $50.2$ | $73 \pm 5$ | $-4.54 \pm 0.08$ | $\mathbf{0.08 \pm 0.01}$ |
| | MEIBP | N/A | N/A | N/A | N/A |
| EEG [12] | FAB | $1.6$ | $\mathbf{11.2 \pm 1.6}$ | $-0.93 \pm 0.02$ | $-0.76 \pm 0.04$ |
| $120576 \times 32$ | EM | $3.7$ | $32 \pm 0$ | $\mathbf{-0.88 \pm 0.09}$ | $-0.59 \pm 0.01$ |
| | IBP | $53.0$ | $46.4 \pm 4.4$ | $-3.16 \pm 0.03$ | $\mathbf{-0.26 \pm 0.05}$ |
| | MEIBP | N/A | N/A | N/A | N/A |
| Piano [21] | FAB | $19.4$ | $58.0 \pm 3.5$ | $\mathbf{-0.83 \pm 0.01}$ | $-0.63 \pm 0.02$ |
| $57931 \times 161$ | EM | $50.1$ | $158.6 \pm 3.4$ | $\mathbf{-0.82 \pm 0.02}$ | $\mathbf{-0.45 \pm 0.01}$ |
| | IBP | $55.8$ | $89.6 \pm 4.2$ | $-1.83 \pm 0.02$ | $-0.84 \pm 0.05$ |
| | MEIBP | $\mathbf{14.3}$ | $\mathbf{48.4 \pm 3.2}$ | $-7.14 \pm 0.52$ | $-6.90 \pm 0.50$ |
| yaleB [7] | FAB | $\mathbf{2.2}$ | $\mathbf{77.2 \pm 7.9}$ | $\mathbf{-0.37 \pm 0.02}$ | $-0.29 \pm 0.03$ |
| $2414 \times 1024$ | EM | $50.9$ | $929 \pm 20$ | $-4.60 \pm 1.20$ | $\mathbf{0.80 \pm 0.27}$ |
| | IBP | $51.7$ | $94.2 \pm 7.5$ | $-0.54 \pm 0.02$ | $-0.35 \pm 0.02$ |
| | *MEIBP | $7.2$ | $69.8 \pm 2.7$ | $-1.18 \pm 0.02$ | $-1.12 \pm 0.02$ |
| USPS [13] | FAB | $\mathbf{11.2}$ | $\mathbf{110.2 \pm 5.1}$ | $\mathbf{-0.96 \pm 0.01}$ | $-0.64 \pm 0.02$ |
| $110000 \times 256$ | EM | $45.7$ | $256 \pm 0$ | $-1.06 \pm 0.01$ | $\mathbf{-0.36 \pm 0.01}$ |
| | IBP | $61.6$ | $181.0 \pm 4.8$ | $-2.59 \pm 0.08$ | $-0.76 \pm 0.01$ |
| | *MEIBP | $1.9$ | $22.0 \pm 2.7$ | $-1.35 \pm 0.03$ | $-1.31 \pm 0.03$ |

this is reasonable, for large $N$, both of them ideally achieve the maximum likelihood while *FAB* achieved much smaller $K$ (see identifiability discussion in Section 3). In small $N$ scenarios, on the other hand, FIC approximation would be not accurate, and *FAB* would perform worse than NPBs (while we observed such case only in Libras.)

## 5 Summary

We have considered here an FAB framework for LFMs that offers fully automated model selection, i.e., selecting the number of latent features. While LFMs do not satisfy the assumptions that naturally induce FIC/FAB on MMs, we have shown that they have the same "degree" of model complexity as the approximated marginal log-likelihood, and we have derived FIC/FAB in a form similar to that for MMs. In addition, our proposed accelerating mechanism for shrinking models drastically reduces total computational time. Experimental comparisons of FAB inference with existing methods, including state-of-the-art IBP methods, have demonstrated the superiority of FAB/LFM.

## Acknowledgments

The authors would like to thank Finale Doshi-Velez for providing Piano and EEG data sets. This work was supported by JSPS KAKENHI Grant Number 25880028.

## Footnotes

[1] While $p(\mathbf{X}|\mathcal{P})$ is a non-regular model, $P(\mathbf{X}, \mathbf{Z}|\mathcal{P})$ is a regular model (i.e., the Fisher information is non-singular at the ML estimator,) and $\mathbf{F}_k$ and $\mathbf{F}_Z$ have their inversions at $\hat{\mathcal{P}}$.

[2]We also investigated the effect of merge post-processing, but none was observed in this small example.

[3]We totally omitted *VB* because of its long computational time.

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
