[Supplementary Material]

# Supplementary Materials for "Factorized Asymptotic Bayesian Inference for Latent Feature Models"

## A    Lower Bound and Update Equations in M-step

The expectations in lower bound (10) are explicitly written as:

$$\mathbb{E}_q\left[\log p(\mathbf{X}|\mathbf{Z},\boldsymbol{\Theta})\right] = -\frac{1}{2}\mathrm{tr}(\boldsymbol{\Lambda}\mathbf{W}\sum_n \mathbb{E}_q[\mathbf{z}_n\mathbf{z}_n^\top]\mathbf{W}^\top - 2\boldsymbol{\Lambda}\mathbf{W}\sum_n \boldsymbol{\mu}_n\bar{\mathbf{x}}_n^\top + \boldsymbol{\Lambda}\sum_n \bar{\mathbf{x}}_n\bar{\mathbf{x}}_n^\top)$$
$$+ \frac{N}{2}\sum_d \log \lambda_d, \tag{15}$$

$$\mathbb{E}_q[\log p(\mathbf{Z}|\boldsymbol{\pi})] = \boldsymbol{\eta}(\boldsymbol{\pi})^\top \sum_n \boldsymbol{\mu}_n - N\sum_k \log(1 + e^{\eta(\pi_k)}). \tag{16}$$

By taking gradients of lower bound (10) with respect to $\mathcal{P}$, we obtain the following closed-form solutions:

$$\mathbf{W} = \sum_n \bar{\mathbf{x}}_n\boldsymbol{\mu}_n^\top \left(\sum_n \mathbb{E}_q[\mathbf{z}_n\mathbf{z}_n^\top]\right)^{-1}, \tag{17}$$

$$\frac{1}{\lambda_d} = \frac{\sum_n(\mathbf{w}_d^\top\mathbb{E}_q[\mathbf{z}_n\mathbf{z}_n^\top]\mathbf{w}_d - 2\bar{x}_{nd}\boldsymbol{\mu}_n^\top\mathbf{w}_d + \bar{x}_{nd}^2)}{N}, \tag{18}$$

$$\mathbf{b} = \frac{\sum_n(\mathbf{W}\boldsymbol{\mu}_n - \mathbf{x}_n)}{N}. \tag{19}$$

## B    The Algorithm of `merge`

Algorithm 1 summarizes the procedures of `merge`, which is used in line 10 of Algorithm 1.

---
**Algorithm 1** `merge`

---
**input** $\{\boldsymbol{\mu}_n\}, \mathbf{W}$
1: **for** $k = 1, \ldots, K$ **do**
2:     Create distance matrix $\mathbf{D}$ from $\mathbf{W}$
3:     $l \leftarrow \mathrm{argmin}_{k'} D_{kk'}$
4:     $\mathbf{w}'_{\cdot k} \leftarrow 2\mathbf{w}_{\cdot k}, \mathbf{w}'_{\cdot l} \leftarrow \mathbf{0}$
5:     $\boldsymbol{\mu}'_{\cdot k} \leftarrow (\boldsymbol{\mu}_{\cdot k} + \boldsymbol{\mu}_{\cdot l})/2, \boldsymbol{\mu}'_{\cdot l} \leftarrow \mathbf{0}$
6:     **if** $\mathcal{L}(\mathbf{W}', \{\boldsymbol{\mu}'_n\}) \geq \mathcal{L}(\mathbf{W}, \{\boldsymbol{\mu}_n\})$ **then** $\mathbf{W} \leftarrow \mathbf{W}', \{\boldsymbol{\mu}_n\} \leftarrow \{\boldsymbol{\mu}'_n\}$
7: **end for**

---

## C    Shrinkage Acceleration and Exponentiated Gradient

The optimization procedure of shrinkage acceleration discussed in Section 3 has another interpretation as the exponetiated gradient descent [1], which is a gradient-based optimization technique on a simplex. The exponetiated gradient algorithm maximizes Eq. (13) by iteratively solving the following updates:

$$q^t(\mathbf{Z}) = \mathrm{argmax}_q \mathcal{L}(q) + \alpha \sum_{\mathbf{Z}} q(\mathbf{Z})\log\frac{q(\mathbf{Z})}{q^{t-1}(\mathbf{Z})}, \tag{20}$$

where the second term is a proximal one of the KL-divergence from the previous solution. By taking the derivative to zero, the solution is given as

$$q^t(\mathbf{Z}) \propto (q^{t-1})^{1-\frac{1}{\alpha}}\exp(\alpha\nabla\mathcal{L}(q^{t-1})). \tag{21}$$

(a) $N = 500$.

(b) $N = 2000$.

(c) $N = 5000$.

Figure 1: Estimated $K$ v.s. elapsed time over 10 trials (left) and learned $\mathbf{W}$ (right) in `block` data.

In contrast with the CCCP, the exponentiated gradient has a tunable step size $\alpha > 0$. If we set a small $0 < \alpha < 1$, the solution moves faster. We can easily confirm that solution (21) is equivalent to that of the CCCP (14) with $\alpha = 1$. This property may reduce computational cost in our acceleration scheme; currently `accelerateShrinkage()` requires hundred iterations to obtain a sparse solution of $\{\boldsymbol{\mu}_n\}$, but that procedure is potentially replaced by a few iteration with small $\alpha$.

## D  Additional Block Data Experiments

Figure 1 depicts the relationship between estimated $K$ and elapsed time (left panel) and examples of estimated $\mathbf{W}$ (right panel) in which TLL is the median over 10 trials. We observe that, the larger the number of samples $N$, the more accurate the model selection of *FAB*. This is reasonable because of FIC's asymptotic property. Indeed, for $N = 5000$, *FAB* correctly selected $K$ at 8 trials out of 10.

# E   Proofs

First, let us summarize the assumptions we will use for the following proofs: **A1)** the prior of $\mathcal{P}$ can be factorized as: $p(\mathcal{P}|\mathcal{M}) = p(\boldsymbol{\pi}|\mathcal{M}) \prod_d p(\boldsymbol{\theta}_d|\mathcal{M})$, **A2)** those priors are continuous, and **A3)** the log-priors are constant with respect to $N$, i.e., $\lim_{N\to\infty} \frac{\log p(\boldsymbol{\theta}_d|\mathcal{M})}{N} = 0$. Further, when we consider the asymptotic behaviour, let us assume $\mathbf{z}_n$ for $n = 1, \ldots, N$ to be independent random variables such that: **A4)** for sufficiently large $N$, $\sum_n z_{nk}/N$ converges in probability to $p_k$ such that $0 < p_k < 1$.

Before giving the proof of Lemma 1, let us introduce the following definition and lemmas.

**Definition 5.** *The exponential family representation of the Gaussian likelihood $p(x_n|\mathbf{z}_n, \boldsymbol{\theta})$ has the natural parameter $\boldsymbol{\xi}_n = (\lambda(\mathbf{w}^T\mathbf{z}_n + b), -\lambda/2)$ and the log-partition function $\psi(\boldsymbol{\xi}) = -\frac{\xi_1^2}{4\xi_2} - \frac{1}{2}\log 2\xi_2$. The negated Hessian matrix of the log-likelihood with respect to $\boldsymbol{\theta}$ is then given as*

$$-\frac{\partial^2}{\partial\boldsymbol{\theta}\partial\boldsymbol{\theta}^\top} \log p(x_n|\mathbf{z}_n, \boldsymbol{\theta}) = \frac{\partial\boldsymbol{\xi}_n}{\partial\boldsymbol{\theta}} \boldsymbol{\Psi}^{(n)} \frac{\partial\boldsymbol{\xi}_n}{\partial\boldsymbol{\theta}}^\top, \tag{22}$$

*where $\boldsymbol{\Psi}^{(n)}$ is the Hessian matrix of $\psi(\cdot)$ in which the elements are given by $\Psi_{11}^{(n)} = \frac{-1}{2\xi_2}$, $\Psi_{12}^{(n)} = \Psi_{21}^{(n)} = \frac{\xi_1}{2\xi_2^2}$, and $\Psi_{22}^{(n)} = \frac{\xi_2 - 2\xi_{n1}}{4\xi_2^3}$ (for the sake of clarity, we omitted $n$ from $\xi_{n2}$ because it does not depend on $n$.)*

**Lemma 6.** *For a symmetric, block matrix $\mathbf{M} = \left(\begin{smallmatrix} \mathbf{A} & \mathbf{B} \\ \mathbf{B}^\top & \mathbf{C} \end{smallmatrix}\right)$ is positive definite (PD) if and only if (i) $\mathbf{C}$ and its Schur complement $\mathbf{A} - \mathbf{B}\mathbf{C}^{-1}\mathbf{B}^\top$ are both PD and (ii) $\mathbf{A}$ and its Schur complement $\mathbf{C} - \mathbf{B}^\top\mathbf{A}^{-1}\mathbf{B}$ are both PD.*

**Lemma 7.** *Under A4, $\mathbf{F}^{(d)}$ in Eq. (6) converges in probability to a PD matrix for $N \to \infty$.*

*Proof of Lemma 7.* In this proof, we omit the index $d$ for the sake of clarity. First, according to Eq. (22), $\mathbf{F}$ is rewritten as $\mathbf{F} = \frac{1}{N}\left(\begin{smallmatrix} \mathbf{S} & \mathbf{u} \\ \mathbf{u}^\top & v \end{smallmatrix}\right)$ where

$$\mathbf{S} = \lambda^2 \Psi_{11}^{(n)} \begin{pmatrix} \sum_n \mathbf{z}_n \mathbf{z}_n^\top & \sum_n \mathbf{z}_n \\ \sum_n \mathbf{z}_n & \sum_n 1 \end{pmatrix} /N,$$

$\mathbf{u} = \lambda\Psi_{11}^{(n)}(\sum_n(\mathbf{w}^\top\mathbf{z}_n + b)\mathbf{z}_n/N, \quad \sum_n(\mathbf{w}^\top\mathbf{z}_n + b)/N)$, and $v = \Psi_{11}^{(n)}\sum_n\{(\mathbf{w}^\top\mathbf{z}_n + b)^2 - \frac{\Psi_{12}^2 + \Psi_{11}\Psi_{22}}{4\Psi_{11}^2}\}/N$. By introducing the diagonal matrix

$$\mathbf{D} \equiv \begin{pmatrix} \mathrm{diag}(\sum_n \mathbf{z}_n/N) & \mathbf{0} \\ \mathbf{0} & 1 \end{pmatrix}, \tag{23}$$

we are able to rescale $\mathbf{F}$ as $\mathbf{F} = \mathbf{D}^{1/2}\tilde{\mathbf{F}}\mathbf{D}^{1/2}$. Note that $\mathbf{D}$ is PD because of **A4**, and $\mathbf{F}$ is PD if and only if $\tilde{\mathbf{F}}$ is PD because the product of PD matrices is PD. Thus now we only need to say $\tilde{\mathbf{F}}$ is PD.

Next, let us consider the asymptotic behavior of $\tilde{\mathbf{F}}$. According to **A4**, $\tilde{\mathbf{S}}$ converges in probability as:

$$\tilde{\mathbf{S}} \to \lambda^2\Psi_{11}^{(n)}\begin{pmatrix} \sqrt{\mathbf{p}}\sqrt{\mathbf{p}}^\top + \mathrm{diag}(\mathbf{1} - \mathbf{p}) & \sqrt{\mathbf{p}} \\ \sqrt{\mathbf{p}}^\top & 1 \end{pmatrix}. \tag{24}$$

Since the Schur complement $\lambda(\sqrt{\mathbf{p}}\sqrt{\mathbf{p}}^\top + \mathrm{diag}(\mathbf{1}-\mathbf{p})) - \lambda\sqrt{\mathbf{p}}\sqrt{\mathbf{p}}^\top = \lambda\mathrm{diag}(\mathbf{1}-\mathbf{p})$ is PD by **A4**, Lemma 6 results that $\tilde{\mathbf{S}}$ converges to a PD matrix. Similarly, $\tilde{\mathbf{u}} \to \lambda\Psi_{11}^{(n)}\{(\mathbf{w}^\top\mathbf{p} + b)\sqrt{\mathbf{p}} + \mathbf{w} * (1-\mathbf{p}) * \sqrt{\mathbf{p}}\}$ where $*$ denotes Hadamard product and $v$ converges to a positive number, and using Lemma 6 again yields the statement. $\qquad\square$

*Proof of Lemma 1.* By using the diagonal matrix $\mathbf{D}$ defined in Eq. (23), we have

$$\log \det |\mathbf{F}^{(d)}| = \log \det |\mathbf{D}^{1/2}\tilde{\mathbf{F}}^{(d)}\mathbf{D}^{1/2}|$$

$$= \log \det |\tilde{\mathbf{F}}^{(d)}| + \sum_k \log \frac{\sum_n z_{nk}}{N}.$$

From the proof of Lemma 7, since determinants of $\tilde{\mathbf{S}}$ and Schur complements converge to $O_p(1)$ and $\log \det |\tilde{\mathbf{F}}^{(d)}|$ is the product of them, $\log \det |\tilde{\mathbf{F}}^{(d)}|$ is $O_p(1)$. $\qquad\square$

*Proof of Theorem 2.* **A1**, **A2**, and Lemma 7 enable us to apply Laplace's method [2] separately for each data dimension:

$$\log p(\mathbf{X}, \mathbf{Z}|\mathcal{M}) = \sum_d \log \iint p(\mathbf{x}._d, \mathbf{Z}|\boldsymbol{\theta}_d) p(\boldsymbol{\theta}_d|\mathcal{M}) \mathrm{d}\boldsymbol{\pi}\mathrm{d}\boldsymbol{\theta}_d$$

$$\approx \frac{|\mathcal{P}|}{2} \log \frac{2\pi}{N} + \log p(\mathbf{X}, \mathbf{Z}|\hat{\mathcal{P}}) + \log p(\hat{\mathcal{P}}|\mathcal{M})$$

$$- \frac{1}{2} \sum_k \log \frac{\partial^2 - \log p(\mathbf{z}._k|\pi_k)}{\partial \pi_k^2}\bigg|_{\hat{\pi}_k} - \frac{1}{2} \sum_d \log \det |\mathbf{F}^{(d)}|.$$

By substituting the result from Lemma 1 and ignoring asymptotically constant terms, we obtain the statement. $\square$

*Proof of Theorem 4.* Here we assume $\mathbf{w}^*._k = \mathbf{w}^*._l$. From Eq. (17), the relationship $\mathbf{W}^* \sum_n \mathbb{E}_{q^*}[\mathbf{z}_n \mathbf{z}_n^\top] = \sum_n \mathbf{x}_n (\boldsymbol{\mu}_n^*)^\top$ holds. By substituting the equality into $\mathcal{L}$ and taking a derivative with respect to $\mu_{nk}$, we obtain $\mu_{nk}^* = g(\beta_{nk} + \eta(\pi_k^*) - \frac{D}{2N\pi_k^*})$, where $\beta_{nk} = \frac{1}{2}\mathbf{x}_n^\top \boldsymbol{\Lambda}^* \mathbf{w}^*._k$ and $\pi_k^* = \sum_n \mu_{nk}^*/N$. By taking a summation over $n$ and dividing by $\pi_k^*$ on both sides, we have

$$N = \frac{1}{\pi_k^*} \sum_n g(\beta_{nk} + \eta(\pi_k^*) - \frac{D}{2N\pi_k^*})$$

$$= \sum_n \left( \pi_k^* + (1 - \pi_k^*) \exp(-\beta_{nk} + \frac{D}{2N\pi_k^*}) \right)^{-1}$$

$$= \sum_n \frac{\exp(\beta_{nk})}{\pi_k^* \exp(\beta_{nk}) + \gamma_k},$$

where $\gamma_k = (1 - \pi_k^*) \exp(\frac{D}{2N\pi_k^*})$. We have a similar result for $\mu_{nl}^*$ that $N = \sum_n \frac{\exp(\beta_{nk})}{\pi_l^* \exp(\beta_{nk}) + \gamma_l}$ (Note that the assumption $\mathbf{w}^*._k = \mathbf{w}^*._l$ implies $\beta_{nk} = \beta_{nl}$.) By taking the difference between that of $\mu_{nk}^*$ and $\mu_{nl}^*$, we have

$$\sum_n \frac{(\pi_k^* - \pi_l^*) + \gamma_k - \gamma_l}{\exp(-\beta_{nk})(\pi_k^* \exp(\beta_{nk}) + \gamma_k)(\pi_l^* \exp(\beta_{nk}) + \gamma_l)} = 0.$$

From the conditions for the stationary points, the denominator takes a bounded positive value and the equality holds if and only if the numerator takes zero, i.e.,

$$\pi_k^* + (1 - \pi_k^*) \exp(\frac{D}{2N\pi_k^*}) - \{\pi_l^* + (1 - \pi_l^*) \exp(\frac{D}{2N\pi_l^*})\}$$

$$= (1 - \pi_k^*)(\exp(\frac{D}{2N\pi_k^*}) - 1) - (1 - \pi_l^*) \exp(\frac{D}{2N\pi_l^*} - 1) = 0.$$

Since the function $(1 - \pi)(\exp(\frac{D}{2N\pi}) - 1)$ increases strictly monotonically for $\pi \in (0, 1]$, the equality holds if and only if $\pi_k^* = \pi_l^*$. $\square$