[Reviews · NeurIPS 2013]

Submitted by Assigned_Reviewer_1

This paper proposes a novel model selection criterion for binary latent feature models. It is like variational Bayes, except that rather than assuming a factorized posterior over latent variables and parameters, it approximately integrates out the parameters using the BIC. They demonstrate improved held-out likelihood scores compared to several existing IBP implementations.

The proposed approach seems like a reasonable thing to do, and is motivated by a plausible asymptotic argument. The main advantage relative to other methods for IBP inference is that computationally, it corresponds to an EM algorithm with some additional complexity penalties, rather than the more expensive sampling or variational Bayes algorithms.

The technical contributions seem novel but incremental: they are essentially an extension of the FAB work of [3] and [4].

Some parts of the exposition were confusing because the math seems imprecise. In equation (3), the log sum of the z values is infinite if all of the values are zero. Since this has nonzero probability under any nondegenerate variational distribution q, why isn't the FIC score always infinite?

Theorem 2 states that the marginal likelihood "can be asymptotically approximated as" something, but it's not stated what the asymptotic regime is or what the assumptions are. In particular, how is K assumed to behave as N gets larger? Since the BIC is applied to each component individually, the theorem seems to require a fixed finite K, so that the number of activations of every feature would approach infinity. Under the IBP model, the number of components is infinite, and new components would continue to be observed as the amount of data increases, so assuming finite K removes one of the motivations for using the IBP.

More minor points: in section 2, shouldn't p_k(X | z_{\cdot, k}) include only the data points assigned to mixture component k, rather than the entire dataset? In equation (4), there should be an expectation on the left hand side.

In the quantiative results in Table 1, the proposed method achieves higher predictive likelihood scores in less time compared to alternative methods. While FAB finishes faster by orders of magnitude in some cases, it's not clear how to interpret this result because the stopping criterion isn't specified. It seems like an arbitrary decision when to stop the Gibbs sampler, in particular.

The improvements in predictive likelihood are significant, but where does the difference come from? The results would be more convincing if there's evidence that the difference is due to the model selection criterion rather than which algorithms get stuck in worse local optima, since the latter probably depends more on the details of the implementation (initialization strategies, etc.). (The fact that the Gibbs sampler learns an excessively large number of components in some of the datasets suggests that it's a problem with local optima, since in theory it should be exactly computing the likelihood which FAB is approximating.)
Summary: The proposed approach seems to make sense, but is somewhat incremental. Some of the math seems a bit imprecise. The experimental results show improvements in running time and predictive likelihood compared to previous approaches, but as stated above, I don't think enough analysis was done to be able to interpret these results.

Submitted by Assigned_Reviewer_5

Summary:

The authors propose an algorithm/approximation method for efficient model selection for latent feature models by using factorized asymptotic bayesian (FAB) inference and the factorized information criterion (FIC). FAB and FIC are well known for mixture models, where the latter approximates a model's log likelihood via factorization and Laplace approximation for tractable model selection. The contribution is the generalization of FAB to latent feature models, which uses a mean field approximation for inference of the latents and accelerated shrinkage of the global number of features selected.

A feature of the approach is more automaticity of the model selection, with little hand tweaking. Results are shown on synthetic and real data, and the gain in computational efficiency is significant.

Quality:
Presented results involve thorough analysis and the method is evaluated in comparison to a wide range of competing methods on multiple data sets. References are sufficient.

Clarity:
The paper is clearly written and well structured.

Originality/Significance:
Seems to be a logical extension and solid generalization of previous work in the area of model selection in LFMs using FAB inference and FIC. Namely, the using a combination of several approximations into one novel method that works well.

It would be nice to understand when/how these approximations may lead to poor performance, i.e. "push" the method until it breaks, given that there are several approximations (FIC, mean field, shrinkage) playing a roll at the same time.
Summary: The authors propose an algorithm/approximation method for efficient model selection for latent feature models by using factorized asymptotic bayesian (FAB) inference and the factorized information criterion (FIC). Results are shown on synthetic and real data, and the gain in computational efficiency is significant.

Submitted by Assigned_Reviewer_6

Summary:

The paper presents an inference algorithm for the binary latent feature model (LFM) which expresses each observation x_n as a K size binary vector z_n where z_{nk} = 1 means that x_n has the k-th latent feature present. The goal of inference is to learn the z's for all observations (along with the other model parameters). The presented EM based algorithm uses a shrinkage mechasnism to learn the z's as well as infer the number of latent features (K). This is similar in spirit to nonparametric Bayesian latent feature models such as the Indian Buffet Process (IBP). However, the presented algorithm doesn't actually use prior distributions inducing model parsimony but rather uses a model selection criteria - Factorized Asymptotic Bayesian Inference (FAB) - recently introduced in the context of mixture models and HMMs. THe FAB idea is based on expressing the log marginal likelihood in terms of the expected complete log-likelihood plus a model selection term that encourages shrinkage of the number of latent features (by specifying an upper-bound on K and *learning* the correct K during inference). On some large datasets, the FAB is shown to outperform other inference methods including Gibbs sampling, variational inference, and MAP estimation methods for the IBP (in terms of running time and inference quality).

Quality: The technical quality seems reasonably sound and algorithmic details seem correct. The ability of dealing with large datasets and also learning the number of latent features is very appealing.

Clarity: The paper is reasonably clear in the methodology part. However, some of the experimental details are somewhat unclear (discussed below).

Originality: The proposed method is based on recently proposed framework on Factorized Asymptotic Bayesian (FAB) Inference applied for mixture models and HMMs. The application of FAB to latent feature models however is novel.

Significance: The paper is addressing an important problem (efficient inference in latent feature models while also inferring the number of latent features).

Weak Points:

- There is no discussion about the limitations of the proposed algorithm. Are there any cases when the algorithm might perform worse than vanilla Gibbs sampling?

- There is no discussion about the possible difficulties in convergence (given it is an EM like procedure).

- The algorithm is limited to binary latent feature models (can't be applied for factor analysis or probabilistic PCA).

- The experimental evaluation is not thorough enough (and seems flawed at some points; see comments below).

- In the small-data regime (when the asymptotic argument doesn't hold), it is unclear how the algorithm will behave (there should be some discussion or experiments).

- Some other recent works on efficient inference in LFMs has not been discussed in the paper (see comments below).

- The MEIBP algorithm proposed in [22] is applicable only for non-negative linear Gaussian models (W has to be positive, not real-valued). Therefore, the artificial simulation and the block data experiments are simply invalid for the MEIBP baseline. If you were to compare with MEIBP then the symthetic dataset should have been generated such that the loading matrix W is positive.


Other Comments:

- For block images data, since the noise value is known (and given to the FAB algorithm), for fairness the IBP Gibbs sampler should also be given the same value (instead of the 0.75 std-dev heuristic).

- For VB-IBP [2], the infinite variational version could be used (the experiments used finite variational version). The experimental settings for VB isn't described in enough details (e.g., how many restarts were given?).

-I am surprised that the accelerated Gibbs sampling discovered about 70 latent features on the 49 dimensional Sonar data!). I suspect it is because of bad initialization and/or badly specified noise variance value.

- Line 70: The reference [21] isn't about Gibbs sampling but rather MAP estimate for the IBP (just like reference [22]). Please modify the text and correct the description.

- There is recent work on efficient inference using small-variance asymptotic in case of nonparametric LFMs (see "MAD-Bayes: MAP-based Asymptotic Derivations from Bayes" from ICML 2013). It would be nice to discuss this work as well.

Minor comments:

- Line 349: For real-data, the suggestion in [1,22] was not to set std-dev equal to 0.75, but to set it equal to 0.25 *times* the std-dev of examples across all the dimensions.

*********************************************
Comments after the author-feedback: The feedback answered some of the questions. There are a few things that need to be fixed/improved before the paper could be accepted. In particular:
- Experimental methodology (for the proposed method and the baseline) need to be better explained, and there should be justifications/explanations about why certain algorithms behaved in a certain way (e.g., Gibbs sampler used a fixed alpha that might have led to an overestimated K, or noise variance hyperparamters weren't properly set).
- Reference [21] does MAP inference, not sampling. Please fix this.
- Include the reference on MAD-Bayes inference for the IBP.
Summary: The proposed inference method for LFMs is interesting and the fact that it scales to large datasets (in addition to inferring K) is appealing. However, the experimental evaluation should have been more carefully done.

Submitted by Assigned_Reviewer_7

In this paper, the authors extend factorized asymptotic Bayesian (FAB) inference for latent feature models. FAB is a recently-developed model selection method with really good results for mixture models (MMs) and hidden Markov models. In short, this method maximizes a lower bound of a factorized information criterion (FIC) which converges to marginal log-likelihood. The limitation of the FAB is that it has only been applicable to models satisfying the condition that the Hessian matrix of a complete likelihood should be block diagonal. The authors extend FAB to latent feature models (LFMs) despite the fact the condition is not satisfied. They effectively derive a lower bound of FIC for LFMS and show that it has the same representation ad the FIC for MMs. They provide results on both synthetic and real world datasets and compare FAB/LFM to other methods such as fast GIbbs sampling in models that use Indian Bufffet process, models that use variational Bayes (VB) and maximum-expectation IBP (MEIBP). The results illustrate the superiority of FAB?LFM; the proposed method claimed better performance not only in the prediction task but also in terms of computational cost.

The paper is well written and the authors thoroughly describe the derivation steps of their method.

The proposed model extends FAB/MMs to FAB/LFMs and would consider it incremental. It is quite interesting and the results underline it's efficiency. Although I am not an expert on this, I think that it would be of great interest to the NIPS community.
Summary: All in all, it is a nice paper that presents a really interesting idea. The results are convincing and I support its acceptance.
Author Feedback

Author rebuttal: Thank you for giving us a lot of insightful and helpful discussions.

Before starting our response, we would like to clarify that we do not intend to claim finite LFMs are better than infinite LFMs in general. Of course, as reviewers pointed, infinite ones does not require the ``finite K'' assumption. Our observation is that, as a combination of model and algorithm, finite LFM + FAB achieved better predictive accuracy with less computational cost than infinite LFMs (IBPs) + existing inference methods (Gibbs, VB).


[For Assigned_Reviewer_1]

> In equation (3), the log sum of the z values is infinite ...

If z_nk=0 for n=1,...,N, w_dk does not appear in the likelihood p(X|Z,Theta), and the marginalization w.r.t. w_dk does not affect to the (true) marginal likelihood (see 651 of the supplementary material). Without loss of generality, we can marginalize out such w_dk before applying Laplace's method, which is equivalent to remove z_k from the log sum z term, and FIC does not go infinity.


> Theorem 2 states that ..., but it's not stated what the asymptotic regime is or what the assumptions are.

As described at the beginning, we assume that K is finite regardless of N. While this assumption is different from IBP, We believe there are many situations that our assumption is valid in the real world (at least the real data sets we evaluated).


> ... it's not clear how to interpret this result (Table 1) because the stopping criterion isn't specified.

As stopping criteria, we basically followed the original papers and we believe this setting is fair: [Gibbs] Stopped after 500 iterations (same setting for real data in [1]). [VB] Stopped when the improvement of the lower bound was less than the threshold. [MEIBP] Same as VB (we set its threshold as 10^-4 as used in [22].) In addition, for the real data, we forced to stop all the methods when the runtime exceeded 50 hours (see 374).

> The improvements in predictive likelihood are significant, but where does the difference come from?

One plausible reason is the difference of the regularizations. Experimental results show FAB could better mitigate over-fitting (TLL and PLL were close) while IBP was likely to over-fit particularly to small datasets.


[For Assigned_Reviewer_6]

> Are there any cases when the algorithm might perform worse than vanilla Gibbs sampling?

In principle, LFM/FAB does not have ``tunable'' hyperparameters because of its asymptotic prior ignorance. Since LFM/IBP has a few hyper-parameters and therefore it could outperform FAB in the scenario where domain knowledge can be represented as IBP prior (i.e., those hyper-parameters can be determined on the basis of domain knowledge).

> There is no discussion about the possible difficulties in convergence (given it is an EM like procedure).

The FAB E- and M-step monotonically maximize the FIC lower bound as described in Theorem 6 of [5] and Theorem 2 of [4], and the parameters converge in local maxima after sufficiently large number of iterations.

> In the small-data regime (when the asymptotic argument doesn't hold), it is unclear how the algorithm will behave

Since FAB is an asymptotic method, it's important issue to investigate its behavior over N. Fig.2 shows that FAB worked fairly well in relatively small sample cases (N=100, 200, ..). We observed the same tendency in Sonar data.

> For block images data, ...

We're afraid that this point is a reviewer's misunderstanding. FAB learns data variance (see Eq.18) and we did not specify the true noise variance for all experiments. Since the Gibbs sampling method does not learn the variance (at least in its software), we specified it by the 0.75 heuristic by following [22]. We believe this experimental setting was fair.

It is worth noting that we specified the true variance for IBP methods in the artificial simulations except the block data (see 348) but still FAB performed better.


> The MEIBP algorithm ...

We agree that some experiments were inappropriate for MEIBP (we did not notice that MEIBP is for non-negative data only.) We fix this in the final manuscript. Nevertheless, we believe the other experiments including real world datasets show the superiority of FAB over the other methods.

> The experimental settings for VB isn't described in enough details.

We set alpha (hyperparameter of IBP prior) as 3 and sigma_n and sigma_z as 0.75 by following [22]. For other parameters, we followed the default setting of tester.m in http://people.csail.mit.edu/finale/code/ibp-gibbs.tar, e.g. # random restart of VB is 5. We omitted the results of infinite VB because of the space limitation and its performance was roughly same as the finite one.

> I am surprised that the accelerated Gibbs sampling discovered about 70 latent features ...

As you mentioned, in a few data sets (Sonar, Auslan, and EEG), estimated K of IBP was larger than D, but TLL was the highest, which implies the model was correctly learned. This fact seems not to support that there were bad initialization and/or wrong specification of the variance. We guess this is because the model regularization of IBP did not work well (note that we set alpha as 3 by following [22].)

> MAD-Bayes

Thank you for the suggestion. We did not know the work. We will add a discussion with this work in a final manuscript.

> the suggestion in [1,22] was not to set std-dev equal to 0.75, but to set it equal to 0.25 *times* the std-dev of examples ...

As described in 319, we firstly normalized all the data (std of the resulting data is 1) and then set the noise std as 0.75, and it was almost equivalent to set 0.75 * std. We also tested std=0.25, but std=0.75 (this setting is described in [22]) was overall better in terms of PLL so we employed this setting.